# Non-Invasive vs. Invasive Markers in Ulcerative Colitis: A Systematic Review of Intestinal Ultrasound, Biopsy, and Faecal Calprotectin

**DOI:** 10.3390/ijms26178129

**Published:** 2025-08-22

**Authors:** Viviana Parra-Izquierdo, Juliette De Avila, Oscar Gómez, Nelson Barrero, Miguel Duarte, Consuelo Romero-Sánchez

**Affiliations:** 1Cellular and Molecular Immunology Group—InmuBo, Universidad El Bosque, Bogotá 111321, Colombia; dravivianaparra@gmail.com (V.P.-I.); dequiroga@unbosque.edu.co (J.D.A.); osgomezc@gmail.com (O.G.); 2Gastroenterology Department, Hospital Internacional de Colombia, Piedecuesta 681011, Colombia; 3Gastroadvanced IPS, Bogotá 111311, Colombia; 4Rheumatology and Immunology Department, Hospital Militar Central, Bogotá 110231, Colombia; est.nelson.barrero@unimilitar.edu.co; 5Dermatology Department, Hospital Militar Central, Bogotá 110231, Colombia; dermatologiamiguelduarte@gmail.com; 6Clinical Immunology Group, School of Medicine Hospital Militar Central, Universidad Militar Nueva Granada, Bogotá 110231, Colombia

**Keywords:** intestinal ultrasound, fecal calprotectin, ulcerative colitis, histological remission

## Abstract

Accurate assessment of histological remission is a critical goal in the management of ulcerative colitis (UC); however, routine evaluation is hindered by the invasiveness of endoscopy and biopsy. Non-invasive alternatives like intestinal ultrasound (IUS) and faecal calprotectin (FC) show promise for monitoring mucosal inflammation, though their ability to predict histological healing remains underexplored. This systematic review and meta-analysis aimed to evaluate the diagnostic accuracy of IUS, FC, and their combined use for detecting histologic remission in patients with UC. A comprehensive literature search identified two eligible studies comprising 72 patients. Pooled estimates for IUS demonstrated high sensitivity (0.84, 95% CI: 0.35–0.98) but variable specificity (0.78, 95% CI: 0.08–0.99), while FC alone exhibited high sensitivity (0.85, 95% CI: 0.72–0.92) with moderate specificity (0.60, 95% CI: 0.38–0.79). Notably, only one study assessed the combined diagnostic approach, reporting superior performance with sensitivity and specificity of 0.88 and 0.80, respectively. The certainty of the evidence was rated as moderate. These exploratory findings suggest that a multimodal, non-invasive approach combining IUS and FC may improve diagnostic accuracy in detecting histological remission in UC, potentially reducing reliance on invasive procedures. However, given the limited number of studies included and the high degree of heterogeneity, these results should be interpreted with caution. Further large-scale, methodologically robust studies are needed to validate these preliminary findings and establish standardized diagnostic protocols.

## 1. Introduction

Ulcerative colitis (UC) is a chronic inflammatory bowel disease (IBD) characterized by a relapsing and remitting course, where accurate assessment of inflammatory activity is essential to guide therapeutic decisions and predict long-term outcomes. In recent years, the treatment paradigm has shifted from targeting only clinical and endoscopic remission to also achieving histological remission, a goal increasingly valued due to its association with lower rates of relapse, hospitalization, colorectal cancer, and colectomy.

However, histological assessment requires invasive procedures such as colonoscopy, which limits its practicality for frequent monitoring of disease activity in patients with established ulcerative colitis. This has led to growing interest in non-invasive tools with potential prognostic value capable of reflecting histological remission and guiding treatment decisions over time. Among these, intestinal ultrasound (IUS) and faecal calprotectin (FC) have demonstrated promising correlations with histologic inflammation and are being explored as surrogate markers for mucosal healing. Nevertheless, their ability to accurately reflect histologic remission across different clinical scenarios remains under active investigation.

IUS, in particular, has gained attention as a non-invasive, repeatable, and real-time modality for assessing inflammatory activity and treatment response in UC. Its prognostic value is especially relevant in contexts where colonoscopy is less feasible—such as during pregnancy or in patients with significant comorbidities. IUS is increasingly integrated into routine monitoring strategies for inflammatory bowel disease (IBD), contributing to more dynamic and patient-centered therapeutic decision-making [1]. The main variables assessed using IUS include bowel wall thickness (BWT), presence of color Doppler signal (CDS) as an indicator of hyperemia, stratification of bowel wall layers, inflammatory mesenteric fat, mesenteric lymphadenopathies, and, in specific cases, complications such as strictures or abscesses. Among these, BWT is the most relevant parameter, as it correlates significantly with both endoscopic and histologic activity; values < 2.8–3 mm are typically associated with endoscopic and histologic remission [2,3].

Conversely, calprotectin, which is alternatively referred to as MRP8/14 and S100A8/A9, is a calcium- and zinc-binding protein belonging to the S-100 protein family, comprising over 24 members in vertebrates, and exhibits evolutionary conservation, having been initially isolated from blood leukocytes [4,5]. It exists as an oligomer composed of two light subunits (11 kDa each) and one heavy subunit (13 kDa), culminating in a total molecular mass of approximately 36.5 kDa [5]. The genetic sequences encoding the calprotectin subunits are designated as S100A8 and S100A9, situated on chromosome 1q21 [6]. MRP8 is regarded as the functional subunit, while MRP14 serves to inhibit the premature degradation of MRP8 [4,7].

Calprotectin constitutes 60% of cytosolic proteins in neutrophils, with lesser presence in monocytes and macrophages, found in various bodily fluids [6]. It is involved in diverse physiological processes such as cell differentiation, immune regulation, tumorigenesis, apoptosis, and inflammation [4,5]. Calprotectin is a key player in inflammation and is recognized as a positive acute-phase protein [8]. It enhances receptor expression for neutrophil migration, adhesion, and phagocytosis (e.g., CD35, CD66b, CD18, CD11b), facilitates chemotaxis, and functions as a damage-associated molecular pattern protein in innate immunity. The S100A8/S100A9 complex controls intracellular pathways of innate immune cells and allows orchestration of an inflammatory response [8]. Calprotectin modulates cytoskeletal rearrangements to allow leucocyte recruitment and facilitates the transport of arachidonic acid to sites of inflammation [9].

In IBD, but also in many other inflammatory conditions, clinicians increasingly employ FC, a well-studied (systemic and faecal) inflammatory biomarker due to its stability, assay reproducibility, and low cost to guide diagnostic and therapeutic decisions [9]. FC can discriminate between non-inflammatory and inflammatory disease of the intestine, can be retrieved non-invasively, and remains stable at room temperature in stool for at least 3 days (with 30% inadequacy after 7 days) [10]. Several studies suggested that FC allowed differentiation between non-IBD and IBD at a cut-off between 100 and 200 µg/g [11].

The available evidence indicates that FC does show promise as a non-invasive surrogate biomarker for predicting histological remission in UC. Langhorst et al. confirmed that FC moderately correlated with histological scores when histological inflammatory activity was evaluated, suggesting its potential to discriminate between active inflammation and remission states [12]. Clinical decision-making to discriminate inflammatory from non- inflammatory disease with relative accuracy and the need for endoscopy is indicated, however, the European Crohn’s and Colitis Organization (ECCO) position paper also underscores the clinical relevance of FC as its levels strongly correlate with histological indices, even though the exact cut-off values remain a topic for further standardization [13].

Additionally, the meta-analysis by Ye et al. pooled data across multiple studies and reported a sensitivity of around 76% and a specificity of 71% for FC in identifying histological remission, with optimal FC cut-off values generally placed between 100 and 200 µg/g [14]. Park et al., using the Robarts Histopathology Index, further quantified the predictive performance of FC, identifying a cut-off around 80 mg/kg with moderate sensitivity and specificity, thereby reinforcing FC’s practical utility in predicting histologic remission [15]. FC is a reliable predictor of histologic remission, with higher accuracy at lower thresholds. It could help stratify patients’ need for colonoscopy for the assessment of endoscopic and histologic remission [16].

FC alone has also demonstrated a strong correlation with histologic remission in UC. Multiple studies have shown that low FC levels (typically between 60 and 100 μg/g) are associated with histological healing, supporting its use as a non-invasive biomarker in clinical follow-up. For example, low FC levels predicted histologic remission and prolonged clinical remission in patients with endoscopic healing [17] and demonstrated high accuracy in therapeutic outcome evaluation [18]. Other studies have corroborated these findings, highlighting FC thresholds as reliable indicators of mucosal and histological healing [19,20,21,22].

The combination of IUS and FC is emerging as a highly effective strategy for non-invasive assessment of remission in patients with UC. This combined technique yields a more precise indication of mucosal recovery than either assessment individually. The non-invasive characteristics of both present multiple benefits over conventional approaches; they allow for regular monitoring while minimizing risks linked to repetitive endoscopies, including complications from sedation or possible bowel perforation [23]. Additionally, IUS delivers prompt feedback in the clinical setting, aiding in swift decision-making [3,23,24,25]. Furthermore, FC offers an objective biochemical measure of inflammation that is unaffected by the limitations of subjective symptom reporting [26]. Although recent research trends advocate for endoscopic remission evaluation leveraging this combination, the histological remission continues to be misreported, raising a critical question [3,27,28,29,30,31].

Despite promising findings from individual studies, there is currently a lack of comprehensive evidence synthesis evaluating the effectiveness of combining IUS and FC to monitor histologic remission and disease activity over time in patients with known UC. Variability in study design, inclusion criteria, ultrasound scoring systems, and FC cut-off values makes it difficult to draw generalized conclusions from isolated reports. Therefore, a systematic review is urgently needed to assess the prognostic performance of this non-invasive combination, consolidate existing data, and determine pooled estimates of sensitivity, specificity, and predictive values. Such an analysis would provide high-quality evidence to support the implementation of IUS and FC as complementary tools in clinical practice and potentially reduce the reliance on invasive procedures such as colonoscopy for routine monitoring of histologic remission. Based on the above, we proposed to review to evaluate the ability of the combined non-invasive strategy using IUS parameters together with FC levels—to reflect histological remission in patients with UC, using histological assessment of intestinal biopsy as the reference standard.

## 2. Methods

This systematic review with meta-analysis was designed and conducted following the Cochrane Diagnostic Test Accuracy guidelines. Our methods encompass a comprehensive literature search, strict eligibility criteria for study design and population, rigorous data extraction, and quality assessment processes. We focused on studies reporting data that allowed us to evaluate the accuracy of both intestinal ultrasound and FC, either individually or in combination, for predicting histological remission in patients with ulcerative colitis, using histological analysis as the reference standard.

### 2.1. Criteria for Considering Studies for This Review

The following eligibility criteria were established to ensure that relevant studies reporting data on non-invasive approaches for assessing histological remission in patients with UC were included in this systematic review. By restricting the included studies to those that meet these criteria, we aim to generate synthesized evidence that is both clinically relevant and impactful for guiding future research and practice. The inclusion criteria are organized into five principal categories.

#### 2.1.1. Types of Studies

Both prospective and retrospective studies were eligible. We considered studies with a cross-sectional design where participants receive both the index test(s) (IUS and/or FC) and the reference standard (histological analysis) at a single point in time, as well as those with a longitudinal component for assessing the reference standard over time. Where available, comparative diagnostic accuracy studies in which all individuals undergo all index tests were considered, as these designs allow for within-study comparisons. Randomized studies of test accuracy, where participants receiving the reference standard are randomized to different index tests, were also considered for inclusion if they met the other eligibility criteria.

#### 2.1.2. Participants

Studies must provide clear information on how and where participants were recruited. For example, eligible studies may report on consecutive series of new patients from primary care, specialty clinics, or other healthcare settings. This detailed description of the recruitment process is essential to assess the potential for selection bias.

#### 2.1.3. Index Tests

Eligible studies were those specifically designed to evaluate the diagnostic accuracy of non-invasive tests (either singularly or in combination). We included studies that compared the diagnostic performance of index tests against the same reference standard (i.e., histological analysis), evaluated one or more index tests individually, and described within-study comparisons of index tests, which provide direct evidence on the incremental value of combining methods.

#### 2.1.4. Target Conditions

The target condition for this review was histological remission in patients with UC. Histological remission is defined as the absence of inflammatory activity on intestinal biopsy. This condition serves as the critical endpoint, as accurate non-invasive identification of histological remission is essential for guiding treatment decisions and predicting long-term outcomes in UC management.

#### 2.1.5. Reference Standards

The clinical reference standard in this review was the histological analysis of an intestinal biopsy. Histological analysis is considered the gold standard for confirming histological remission in UC, as it allows for the direct assessment of inflammatory activity at the tissue level. Studies included in this review must use histological evaluation as the reference standard to determine the presence or absence of the target condition. Any studies that utilized alternative reference standards, which are commonly used in clinical practice but are considered inadequate for establishing histological remission, were excluded to maintain methodological rigor and comparability across studies.

#### 2.1.6. Exclusion Criteria

We excluded studies that did not provide sufficient data for calculating the operational characteristics (e.g., sensitivity, specificity, positive and negative predictive values) of FC, IUS, or their combined use compared to histological analysis. Studies published in languages other than English, Spanish, Italian, or Portuguese; case reports; and studies where data for patients with UC could not be separated from those with Crohn’s disease, when reported collectively under inflammatory bowel disease.

### 2.2. Search Methods for the Identification of Studies

#### 2.2.1. Electronic Searches

The literature review was based on the systematic identification of relevant sources through the design and implementation of structured search strategies. An objective approach was employed to identify the key components of the research questions, followed by the selection of appropriate terms from the specialized thesauri of the Virtual Health Library (DeCs), MEDLINE (MeSH), and EMBASE (EMTREE). This process was further refined by incorporating commonly used terms identified from abstracts of previously recognized relevant references. These terms were collected both manually and through automated processing using the MeSH on Demand tool. Additional search terms were also identified by testing search strings in the PubMed interface with the PubMed PubReMiner tool and through feedback provided by project investigators and team members.

A repository of search strings was subsequently generated, organized around key thematic axes of interest for the research. The final search strings were run across MEDLINE (PubMed), EMBASE (EMBASE), CINAHL (EBSCO), Cochrane Central Register of Controlled Trials (Ovid), and ProQuest Dissertations & Theses Global (ProQuest).

No language or date restrictions were applied to the searches, and no additional search filters were used. All searches were executed on 5 February 2025. For further transparency, the full search strategies for each database are provided in an appendix to this review, with the verbatim search strategy for one selected database serving as an example of the overall approach.

#### 2.2.2. Searching Other Resources

To ensure a comprehensive and unbiased search, additional resources were systematically explored. These efforts included grey literature identified by searching ProQuest Dissertations & Theses Global (ProQuest), handsearching conducted by carefully reviewing the full texts of studies selected during the title and abstract screening process. Finally, the reference lists of all selected studies and other relevant articles were scrutinized to identify further pertinent studies.

### 2.3. Data Collection and Analysis

#### 2.3.1. Selection of Studies

We structured the selection process into several stages to ensure that only relevant studies were included in the review. First, we downloaded the search results from each database and imported them into Rayyan. In Rayyan, we initially identified potential duplicates using the platform’s automated duplicate detection feature, which were then reviewed and resolved manually. After duplicate removal, we agreed on a standardized screening process and divided the remaining results among pairs of reviewers for title and abstract screening. Any disagreements that arose during this phase were resolved by consensus among the entire review team. Subsequently, studies that passed the initial screening underwent a full-text review, again conducted in pairs. Discrepancies in full-text eligibility were discussed and resolved by team consensus, ensuring that each inclusion and exclusion decision was thoroughly vetted. The final set of selected studies was then reviewed and agreed upon by the whole team before proceeding to data extraction.

A comprehensive total of 1438 records were identified through systematic database searches encompassing MEDLINE, EMBASE, CINAHL, Cochrane Central Register of Controlled Trials, and ProQuest, with an additional record identified through citation tracking. Following the removal of 267 duplicate records, 1161 references were meticulously screened based on their titles and abstracts. Out of these, 1057 records were deemed ineligible and subsequently excluded. A total of 105 full-text reports were assessed for eligibility, although one report could not be successfully obtained. From the 104 remaining reports evaluated for eligibility, ultimately, two studies met the inclusion criteria and were included in the systematic review and meta-analysis (Figure 1). The excluded records and the exclusion reason are described in Appendix A.

#### 2.3.2. Data Extraction and Management

Data extraction was conducted systematically, aiming to capture the key characteristics of each study, such as the study setting, participants’ presentation at recruitment, diagnostic procedures employed (both for the index tests, namely FC and IUS, and the reference standard), as well as other relevant study attributes. To streamline the extraction process and enhance our efficiency, we employed SciSpace, an AI-powered platform designed to assist academics in managing, writing, and publishing their work (See Appendix A). The extracted data were verified independently by two reviewers, ensuring that the information captured was both accurate and comprehensive. Any discrepancies between reviewers were resolved through consensus discussions and, where needed, consultation with the entire review team.

#### 2.3.3. Assessment of Methodological Quality

We used the QUADAS-2 tool to assess the methodological quality of the included diagnostic accuracy studies. Two reviewers independently applied the QUADAS-2 tool to each study, which involved assessing risk of bias and concerns regarding applicability across four key domains: patient selection, index tests, reference standard, and flow and timing. In our evaluation, we adhered closely to the standard QUADAS-2 guidelines, tailoring operational definitions to fit the clinical context of UC. Discrepancies between reviewers were resolved through discussion and, if necessary, by consulting the entire review team to reach consensus. Certainty of evidence was evaluated using the GRADE approach, adapted for diagnostic tests (GRADEpro and Cochrane DTA Handbook).

#### 2.3.4. Statistical Analysis and Data Synthesis

This evaluation was based on two diagnostic accuracy studies included after systematic screening. We extracted true positive (TP), false positive (FP), false negative (FN), and true negative (TN) values from each study based on sample size, sensitivity, and specificity reports, taking into account the Nancy score as the gold standard. Logit transformations of sensitivity and specificity were pooled using an inverse-variance weighted random-effects model only for IUS and FC. Confidence intervals (CIs) were calculated using standard errors derived from binomial proportions. For the accuracy meta-analysis, we used Meta-DiSc version 1.4, a platform designed explicitly for synthesizing data from prognostic test accuracy studies. Pooled estimates were derived using the Der Simonian–Laird random-effects model, in line with the presence of heterogeneity between studies. Statistical heterogeneity was assessed using the I^2^ statistic and Cochran’s Q test. A summary receiver operating characteristic (ROC) curve was constructed using the Moses–Shapiro–Littenberg method to explore potential threshold effects and to visualize the trade-off between sensitivity and specificity across studies. As only two studies met the inclusion criteria, meta-analytic modeling was conducted separately for intestinal ultrasound (IUS) and faecal calprotectin (FC). The combined diagnostic approach (IUS + FC) reported by Goodsall et al. (2024) could not be pooled, as no other study provided comparable composite data [32]. Therefore, statistical synthesis for the combined strategy was not feasible.

## 3. Results

Two studies met the inclusion criteria: Sagami et al. 2020 assessed the diagnostic performance of IUS, focusing on BWT, to predict histological healing in patients with UC. Histologic remission was defined as a Geboes score ≤ 2.0 [33]. Goodsall et al. 2024 [32] evaluated the combined use of IUS parameters (including BWT and CDS with FC) to predict histological remission. Histologic healing was defined as a Nancy Index < 2.0. Both studies used endoscopic biopsies as the reference standard, with IUS performed by trained sonographers [32,33]. In Goodsall et al., FC thresholds of 150 and 250 μg/g were used. Sagami et al. focused solely on sonographic findings. Table 1 shows an overview of the included studies [32,33].

The two studies included in the meta-analysis. Both studies evaluated the diagnostic performance of IUS and, in the case of Goodsall et al., the additional use of FC for detecting histologic remission in UC. Details regarding study design, sample size, ultrasound methodology, reference standards, innovations, validated indices, blinding protocols, and reported diagnostic accuracy are provided [32]. The table highlights important methodological differences and innovations, including the targeted use of transperineal ultrasound by Sagami et al. [33], and the composite diagnostic approach combining IUS and FC implemented by Goodsall et al. 2024 [32].

### 3.1. Risk of Bias

The Quality Assessment of Diagnostic Accuracy Studies (QUADAS)-2 assessment indicated low risk of bias for both studies in patient selection, index test, and reference standard domains. However, Sagami et al. had unclear risk for timing due to an unspecified delay between IUS and histological sampling [33]. Goodsall et al. demonstrated a well-designed study with validated instruments and strong methodological safeguards, supporting the use of composite IUS and FC as a reliable non-invasive marker for histological activity in UC [32]. This study aligns well with treat-to-target strategies and reduces reliance on endoscopy [32]. On the other hand, Sagami et al. introduced transperineal ultrasound as a valuable tool for rectal assessment, addressing limitations in transabdominal access [33]. The technique showed excellent sensitivity but limited specificity, and generalizability is restricted due to the single-center design [32,33].

Figure 2 provides a QUADAS-2 heatmap summarizing the assessment of risk of bias and applicability concerns for the two studies included in the meta-analysis (Sagami et al., 2020 [33]; Goodsall et al., 2024 [32]). The evaluation covers the four QUADAS-2 domains: patient selection, index test, reference standard, and flow/timing. Both studies were rated as having low risk of bias and low applicability concerns across all domains, as indicated by the uniform orange coloring and “Low” labels.

### 3.2. Grading of Recommendations Assessment, Development, and Evaluation (GRADE) Assessment

The overall certainty of evidence, rated using GRADE for diagnostic accuracy, was moderate. This was due to imprecision, stemming from the small number of studies and limited sample size. Risk of bias, indirectness, inconsistency, and publication bias were rated low. A GRADE summary of findings is represented in Table 2. Each domain was evaluated as follows: Risk of Bias: Rated as Low because both included studies used a valid reference standard (histological assessment) and reported blinding between the index test and the reference. Inconsistency: Rated as Low due to similar estimates of diagnostic accuracy across studies, despite differences in ultrasound technique (transperineal vs. Milan Ultrasound Criteria). Indirectness: Rated as Low because the included studies directly assessed UC patients and measured histological outcomes relevant to clinical practice. Imprecision: Rated as Moderate due to small sample sizes and relatively wide confidence intervals, especially for specificity. Publication Bias: Rated as Low, with no missing studies identified and a transparent selection process. Certainty of Evidence: Judged as Moderate, downgraded for imprecision but strengthened by consistency, directness, and low risk of bias.

### 3.3. Diagnostic Performance of Included Studies

Table 3 summarizes diagnostic accuracy measures from the two studies included in the meta-analysis. In the study by Sagami et al., IUS demonstrated a high sensitivity (0.96) but low specificity (0.40), suggesting an excellent ability to detect histologic remission but limited accuracy in ruling out patients not in remission [33]. FC at a threshold of ≥100 µg/g showed slightly lower sensitivity (0.89) but improved specificity (0.53). In contrast, the study by Goodsall et al. reported that IUS had perfect specificity (1.00) but low sensitivity (0.54), indicating a high probability of correctly identifying non-remission but limited detection of remission cases [32]. FC > 100 µg/g improved both sensitivity (0.79) and specificity (0.80) compared to IUS alone. Notably, the combined use of IUS and FC > 100 µg/g further enhanced diagnostic performance, achieving a sensitivity of 0.88, specificity of 0.80, PPV of 0.95, and NPV of 0.57. These findings highlight the potential clinical utility of combining non-invasive biomarkers and imaging modalities to improve the identification of histologic remission in UC, which remains a critical goal for optimizing disease monitoring and treatment strategies.

### 3.4. Pooled Accuracy Estimates

A meta-analysis of the two included studies—Sagami et al. [33] and Goodsall et al. [32]—was conducted to assess the diagnostic accuracy of IUS or FC separately, for predicting histologic remission in UC. The analysis was performed based on pooled estimates, using a random-effects model. The ROC plane illustrates the performance of the individual studies and the pooled estimate, showing that while IUS demonstrates high sensitivity, its specificity is variable (Figure 3). The pooled sensitivity, derived from a random-effects meta-analysis of two studies, was estimated at 0.84 (95% CI: 0.34–0.98), indicating that IUS has a generally good ability to identify patients in histologic remission correctly. However, the wide confidence interval highlights substantial uncertainty, driven mainly by heterogeneity between studies (Table 4). In the sensitivity forest plot, Sagami et al. [33] reported a high sensitivity of 0.96 (95% CI: 0.82–1.00), suggesting excellent detection capability in their cohort. Conversely, Goodsall et al. [32] reported a markedly lower sensitivity of 0.54 (95% CI: 0.33–0.74), underscoring variability in the diagnostic yield of IUS across different settings. Similarly, the pooled specificity was 0.78 (95% CI: 0.08–0.99), reflecting moderate overall specificity but with considerable imprecision (Figure 4). Notably, Sagami et al. reported a low specificity of 0.40 (95% CI: 0.16–0.68) [33], whereas Goodsall et al. documented a perfect specificity of 1.00 (95% CI: 0.48–1.00), highlighting significant inconsistency across studies [32] (Figure 5). These findings suggest that although IUS may be a helpful non-invasive tool for identifying histologic remission, its diagnostic performance is highly dependent on the clinical context, operator expertise, and possibly methodological factors. The wide confidence intervals for both sensitivity and specificity indicate the need for further high-quality, standardized studies to establish the true diagnostic value of IUS in this setting.

This ROC plane plot illustrates the individual and pooled diagnostic performance of IUS for detecting histologic remission. The x-axis represents 1-specificity (false positive rate), and the y-axis represents sensitivity. Each data point corresponds to an individual study, with horizontal and vertical bars indicating 95% confidence intervals for specificity and sensitivity, respectively. The larger circle represents the pooled estimate derived from the meta-analysis. The position of the pooled estimate in the upper left quadrant indicates favorable overall diagnostic performance, with high sensitivity and low false positive rates. However, the wide confidence intervals reflect considerable uncertainty, suggesting heterogeneity between studies and potential limitations in the precision of the pooled estimates.

This forest plot summarizes the sensitivity estimates for IUS in detecting histologic remission from two independent studies, along with the pooled estimate derived using a random-effects model. The point estimate for sensitivity in Sagami et al. [33] was 0.96 (95% CI: 0.82–1.00), indicating excellent ability to identify patients in remission correctly. In contrast, Goodsall et al. reported a lower sensitivity of 0.54 (95% CI: 0.33–0.74), reflecting limited detection capability in that cohort [32]. The pooled sensitivity estimate was 0.84 (95% CI: 0.34–0.98), suggesting an overall high ability of IUS to detect histologic remission, although the wide confidence interval underscores substantial heterogeneity and statistical uncertainty. These findings highlight variability in IUS performance across studies, which may be influenced by factors such as operator expertise, patient population, or disease severity.

This forest plot presents the specificity estimates for IUS in detecting histologic remission, based on data from the same two studies. The specificity reported by Sagami et al. was 0.40 (95% CI: 0.16–0.68), indicating limited ability to identify patients not in remission correctly [33]. In contrast, Goodsall et al. reported a perfect specificity of 1.00 (95% CI: 0.48–1.00), suggesting excellent discrimination in their dataset [32]. The pooled specificity estimate was 0.78 (95% CI: 0.08–0.99), reflecting moderate overall specificity. However, the wide confidence interval again suggests high between-study variability and imprecision in the pooled estimate. These results indicate that, while IUS may offer high sensitivity, its specificity is more variable, potentially limiting its standalone use for confidently ruling out active disease.

The evaluation of FC as a non-invasive biomarker for assessing histologic remission in patients with inflammatory bowel disease demonstrated promising performance, particularly in terms of sensitivity, supporting its role in prognostic monitoring rather than diagnostic application. The ROC plane (Figure 6) illustrates the distribution of study-specific estimates and the pooled diagnostic performance, showing that both included studies cluster within the upper-left quadrant of the ROC space, indicating favorable sensitivity and moderate false positive rates. 

The pooled sensitivity, derived from a random-effects meta-analysis, was 0.85 (95% CI: 0.72–0.92), suggesting that FC reliably identifies patients in histologic remission. This finding was consistent across studies, with Sagami et al. reporting a sensitivity of 0.89 (95% CI: 0.72–0.98) [33] and Goodsall et al. reporting a sensitivity of 0.79 (95% CI: 0.58–0.93), as shown in the sensitivity forest plot [32] (Figure 7).

In contrast, specificity estimates exhibited greater variability and lower overall performance. The pooled specificity was 0.60 (95% CI: 0.38–0.79), as illustrated in the specificity forest plot (Figure 8). Study-specific estimates ranged from 0.53 (95% CI: 0.27–0.79) for Sagami et al. [33] to 0.80 (95% CI: 0.28–0.99) for Goodsall et al., reflecting moderate ability to correctly identify patients not in histologic remission and highlighting the potential for false positive results [32]. 

Table 5 summarizes the univariate analysis for this non-invasive biomarker. These findings suggest that, while FC is a valuable non-invasive tool for monitoring mucosal healing and histologic remission, its moderate specificity limits its use as a standalone confirmatory test. Instead, FC may be more appropriately utilized as a screening or rule-out tool, with positive results warranting confirmation through additional diagnostic modalities such as endoscopy or histology.

## 4. Discussion

This systematic review focused on evaluating the diagnostic accuracy of FC and IUS to predict histological activity in patients with UC. Despite the growing interest in histology as a therapeutic goal, there are very few studies that directly correlate non-invasive methods with histological activity measured by validated indices. This review chose to evaluate the effectiveness of FC in predicting histological activity when combined with IUS, given that it is a widely validated non-invasive biomarker for assessing inflammation in UC, useful not only for predicting endoscopic healing but, more recently, histological healing as well. Low FC levels are significantly associated with endoscopic remission (Mayo 0–1) and especially with histological remission, allowing the discrimination of patients with deep mucosal healing versus those with persistent microscopic inflammation even when the mucosa appears normal endoscopically [17,18,21,34,35].

Several studies have identified cut-off points between 60 and 100 μg/g to predict histological remission, with variable sensitivities and specificities, but generally acceptable for clinical practice [17,18,35]. Therefore, the American Gastroenterological Association recognizes the utility of FC for monitoring inflammatory activity in UC. However, it notes that the certainty of the evidence varies depending on the clinical context and the cut-off used [36,37]. IUS is a non-invasive, innovative, and relevant technique for evaluating mucosal healing, with particular interest in correlating its findings with histological healing in patients with UC, as it allows real-time and repeatable assessment of intestinal inflammation. Key parameters, BWT and CDS, which significantly correlate with endoscopic activity. The use of IUS, particularly with BWT ≤ 3 mm, and absence of hyperemia predicts endoscopic healing with high accuracy [2]. Compared to FC, IUS offers the advantage of identifying the location and extent of inflammation, as well as immediately monitoring therapeutic response segmentally, while FC is a global inflammation marker and can be influenced by other conditions. Compared to endoscopy, IUS is less invasive, more acceptable to patients, does not require bowel preparation or sedation, and enables close and frequent monitoring, although endoscopy remains the standard to confirm mucosal and histological healing [1,3,38].

Therefore, this systematic review identifies two highly relevant studies highlighting the role of FC and IUS, particularly in combination, for monitoring histological remission in UC, supporting a prognostic approach within the treat-to-target strategy and minimizing invasive procedures. Only two studies met the inclusion criteria, underscoring a critical gap in the current literature and the importance of this review.

The first study included, conducted by Goodsall et al., demonstrated that IUS using the Milan Ultrasound Criteria significantly correlates with histological activity measured using the Nancy Histological Index (NHI) (coef. 0.14; *p* = 0.011) [32]. When a composite score combining Intestinal ultrasound activity (MUC > 6.3) and FC (≥100 ug/g) was applied, the strength of the linear association with the NHI increased (coefficient 1.307; 95% CI, 0.43–2.18; *p* = 0.003) [32]. While bowel wall thickness alone showed a statistically significant association, individual Doppler markers did not reach significance. However, the most striking finding was that the non-invasive combination of IUS and FC (≥100 µg/g) markedly improved diagnostic accuracy for detecting active histological inflammation, with 88% sensitivity, 80% specificity, and 95% positive predictive value [32]. This composite strategy outperformed either method alone and stands out as a powerful alternative for clinical monitoring, enabling point-of-care decision making and potentially reducing the need for frequent colonoscopies in UC patients [32].

On the other hand, the study by Sagami et al. introduced transperineal ultrasound (TPUS) as a novel and highly effective tool for evaluating the rectum, a segment where traditional transabdominal ultrasound presents diagnostic limitations [33]. The findings showed that rectal wall thickness measured using TPUS was an independent and more accurate predictor of endoscopic and histological healing than FC alone, with an AUC of up to 0.90 for endoscopic healing and between 0.87 and 0.89 for histological healing [32]. In this study, bowel wall thickness and Limberg score, using TPUS, predicted activity in the Geboes score, Robarts Histopathology Index, and NHI. Sensitivity was high, using 4 mm as the cut-off for bowel wall thickness in all histological indices (e.g., Nancy Histological Index > 1: sensitivity 95.5%, specificity 41.6%, AUC = 0.869 (95% CI: 0.713–0.947), *p* = 0.0011) [32]. Specificity was higher with Limberg score ≥ 2, indicating bowel wall thickening with Doppler signal across all indices (e.g., Nancy Index > 1: sensitivity 59.1%, specificity 76.2%, AUC = 0.812 (95% CI: 0.658–0.907), *p* = 0.0021). FC was also a significant predictor of the Geboes score and Nancy Index (FC ≥ 100 μg/g: e.g., Nancy Index > 1: sensitivity 88.9%, specificity 52.4%, AUC = 0.804 (95% CI: 0.618–0.913), *p* = 0.0419) [33].

In multivariable analysis, bowel wall thickness was the only factor independently associated with lamina propria neutrophils and erosion or ulceration, while FC was independently associated with epithelial neutrophils. Additionally, TPUS allowed clear visualization of rectal wall stratification in all patients, overcoming the anatomical barriers of conventional ultrasound. The combination of TPUS with transabdominal ultrasound provided complete visualization of the colon and rectum, positioning itself as an ideal strategy for implementing the treat-to-target approach in UC. These results not only validate TPUS as a reliable non-invasive marker but also suggest its diagnostic superiority over biomarkers such as FC, especially in assessing rectal activity, a key segment in the clinical and prognostic course of the disease [33].

Both studies highlight a paradigm shift toward the use of non-invasive, accessible, and reproducible techniques that can accurately predict histological activity without the need for biopsies or endoscopy. While FC demonstrated acceptable prognostic performance in the study of Goodsall et al., its clinical utility was significantly enhanced when combined with IUS, reinforcing its role as a complementary biomarker, rather than a standalone monitoring tool [32].

Our exploratory analysis highlights the potential value of combining IUS and FC as a non-invasive strategy to assess histological remission in UC. While FC demonstrated consistently high sensitivity, its specificity remained moderate, limiting its standalone diagnostic utility. Similarly, IUS exhibited favorable sensitivity but variable specificity across studies, further reinforcing the need for complementary diagnostic approaches. Notably, only the study by Goodsall et al. evaluated the diagnostic performance of the combined use of IUS and FC [32]. Their findings revealed that the integration of these two modalities achieved the highest sensitivity (0.88) and specificity (0.80) compared to either test alone, suggesting that a combined diagnostic approach offers superior accuracy for identifying histologic remission [32]. However, because only one study evaluated the composite approach, a meta-analysis was not statistically feasible. The absence of additional studies reporting combined diagnostic performance metrics precluded formal modeling. This highlights a relevant gap in the current literature and underscores the need for further high-quality studies to validate composite non-invasive strategies.

Despite these promising signals, the evidence is constrained by the small number of prospective trials, variable FC thresholds (ranging from 100 to 200 µg/g), and marked between-study heterogeneity. These limitations warrant cautious interpretation, as they may inflate effect estimates and limit generalizability across diverse clinical settings.

To advance toward clinical implementation, future research must prioritize large-scale, multicenter studies with standardized IUS protocols (including machine settings and operator training), uniform FC cut-offs validated against biopsy-based histology, and development of integrated diagnostic algorithms and decision-support tools to streamline non-invasive monitoring in routine practice.

## 5. Conclusions

The results suggest that a multimodal, non-invasive strategy to enhance diagnostic precision in UC may reduce reliance on invasive procedures such as endoscopy. However, further large-scale studies are warranted to validate the additive value of combined IUS and FC assessment and to establish standardized protocols for their implementation in routine clinical practice. This study provides exploratory evidence supporting the diagnostic utility of IUS and FC as non-invasive tools for detecting histologic remission in UC. Notably, the combined use of IUS and FC, evaluated exclusively in the study by Goodsall et al., resulted in superior diagnostic performance, achieving the highest sensitivity and specificity compared to either modality alone [33]. These findings underscore the limitations of relying on a single biomarker or imaging technique for accurately determining histologic remission and highlight the added value of a multimodal, non-invasive approach.

### Recommendations

Integration of IUS and FC in Clinical Practice:

Based on the available evidence, the combined use of IUS and FC should be considered as a complementary strategy to enhance disease monitoring and assessment of histologic activity. This approach may improve clinical decision-making while reducing the need for invasive procedures such as endoscopy.

Standardization of Combined Diagnostic Protocols:

There is a pressing need to establish standardized protocols for the integrated application of IUS and FC, including optimal thresholds for FC and clear criteria for IUS interpretation, to ensure consistency and reproducibility across clinical settings in the prognostic evaluation of disease activity.

Further Large-Scale, High-Quality Studies:

The current body of evidence is limited by heterogeneity, wide confidence intervals, and a scarcity of studies evaluating combined prognostic strategies. Future prospective multicenter studies with rigorous methodological designs are essential to validate these findings and define the optimal role of IUS and FC in monitoring histologic remission.

Incorporation into Treat-to-Target Strategies:

Given the potential of IUS and FC to non-invasively reflect histologic remission, future clinical trials should explore their integration into treat-to-target algorithms aimed at achieving both mucosal and histologic healing, which are recognized as key prognostic indicators in UC.

## Figures and Tables

**Figure 1 ijms-26-08129-f001:**
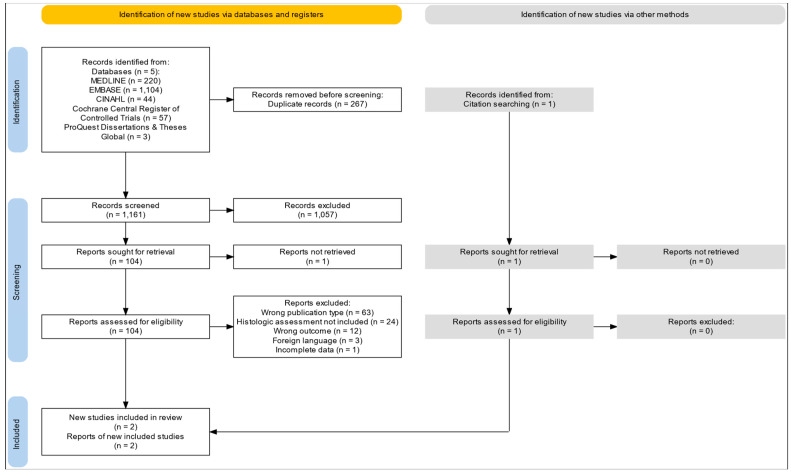
PRISMA 2020 flow diagram illustrating the study selection process. The diagram details the identification, screening, eligibility assessment, and inclusion of studies in the systematic review. Sources include five electronic databases (MEDLINE, EMBASE, CINAHL, Cochrane Central Register of Controlled Trials, and ProQuest Dissertations & Theses Global) and citation searching. Reasons for exclusion are reported at each stage.

**Figure 2 ijms-26-08129-f002:**
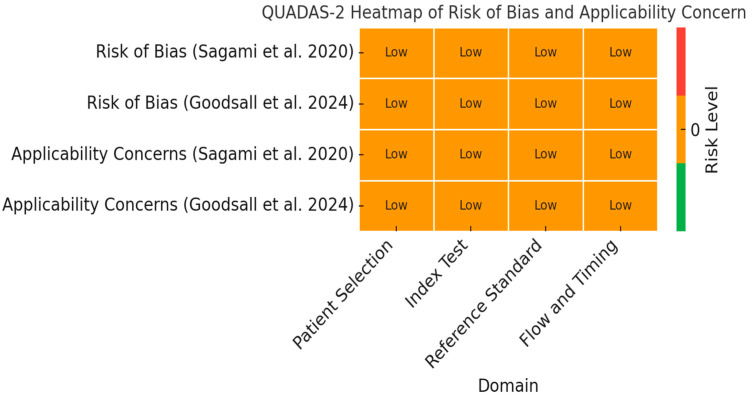
Methodological characteristics of studies evaluating intestinal ultrasound and FC for detecting histologic remission according to QUADAS-2. The color scale represents risk level on a continuous gradient, where green indicates the lowest possible risk (very low), orange represents low risk, and red denotes high risk. The absence of green or red cells confirms that all domains in both studies were judged as low risk [32,33].

**Figure 3 ijms-26-08129-f003:**
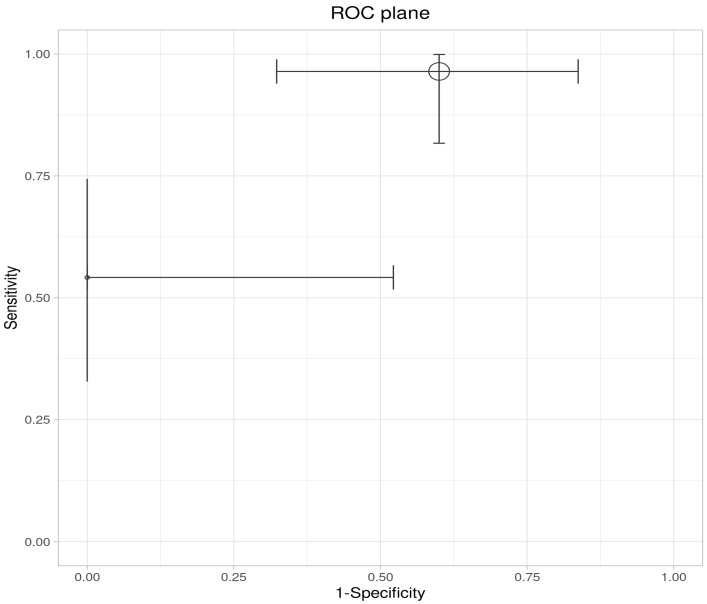
ROC plane for intestinal ultrasound in detecting histologic remission.

**Figure 4 ijms-26-08129-f004:**
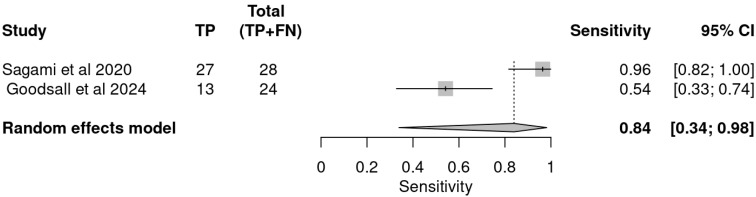
Forest plot of sensitivity estimates for intestinal ultrasound in detecting histologic remission [32,33].

**Figure 5 ijms-26-08129-f005:**
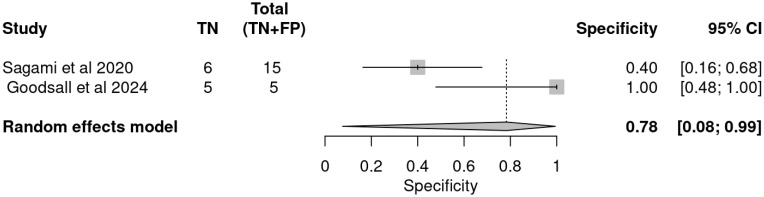
Forest plot of specificity estimates for intestinal ultrasound in detecting histologic remission [32,33].

**Figure 6 ijms-26-08129-f006:**
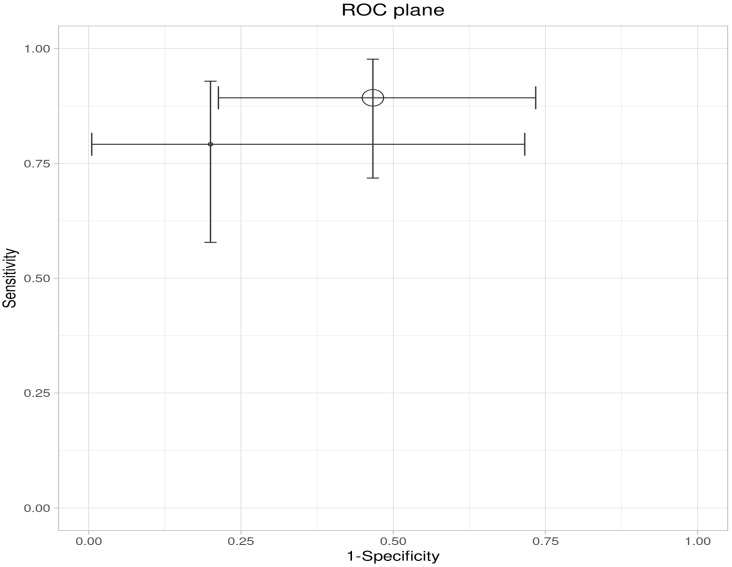
ROC plane for faecal calprotectin in detecting histologic remission.

**Figure 7 ijms-26-08129-f007:**
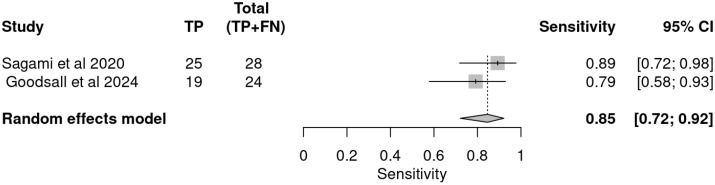
Forest plot of sensitivity estimates for faecal calprotectin in detecting histologic remission [32,33].

**Figure 8 ijms-26-08129-f008:**
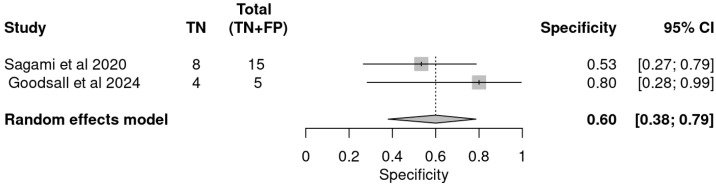
Forest plot of specificity estimates for faecal calprotectin in detecting histologic remission [32,33].

**Table 1 ijms-26-08129-t001:** Comparative overview of study design and methodological features of included studies.

Feature	Goodsall et al. 2024 [32]	Sagami et al. 2020 [33]
**Study Design**	Cohort Study	Cross-sectional diagnostic accuracy study
**Sample Size**	19 patients, 29 assessments	53 patients (43 with histology)
**Ultrasound Method**	Transabdominal (Milan Criteria)	Transabdominal + Transperineal
**Reference Standards**	Nancy Histological Index, Mayo score	Nancy, Robarts, Geboes, Mayo
**Main Innovation**	Composite IUS + FC scoring	Targeted rectal assessment with TPUS
**Validated Indices Used**	MUC, NHI	Nancy, Robarts, Geboes
**Blinding**	Yes, all assessors were blinded	Yes, all assessors were blinded
**Accuracy**	88% sensitivity, 80% specificity	95.5% sensitivity, 41.6% specificity

Table 1 presents a comparative summary of the key methodological features of the two studies included in the meta-analysis. Both studies evaluated the diagnostic performance of IUS and, in the case of Goodsall et al. [32], the use of FC for detecting histologic remission in ulcerative colitis. Details regarding study design, sample size, ultrasound methodology, reference standards, innovations, validated indices, blinding protocols, and reported diagnostic accuracy are provided. The table highlights important methodological differences and innovations, including the targeted use of transperineal ultrasound by Sagami et al. [33], and the composite diagnostic approach combining IUS and FC implemented by Goodsall et al. 2024 [32].

**Table 2 ijms-26-08129-t002:** GRADE assessment of the certainty of evidence for FC and/or intestinal ultrasound in detecting histologic remission.

GRADE Domain	Assessment Details
Outcome	Histological activity detection.
Study Design	Diagnostic accuracy studies (cross-sectional and cohort studies).
Risk of Bias	Low: Studies used blinded histology as a reference standard.
Inconsistency	Low: Similar results across studies despite ultrasound modality differences.
Indirectness	Low: Direct evaluation of UC patients using validated histological indices.
Imprecision	Moderate: Small sample sizes and wide CI in specificity.
Publication Bias	Low: All known relevant studies were included.
Certainty of Evidence	Moderate.

**Table 3 ijms-26-08129-t003:** Diagnostic accuracy of intestinal ultrasound and FC for detecting histologic remission in ulcerative colitis.

Study	Indices	TP	FP	FN	TN	Sensitivity	Specificity	PPV	NVP
**Sagami et al.**	IUS	27	9	1	6	0.96	0.40	0.75	0.86
	FC ≥ 100 µg/g	25	7	3	8	0.89	0.53	0.78	0.73
**Goodsall et al.**	IUS	13	0	11	5	0.54	1.00	1.00	0.31
	FC > 100 µg/g	19	1	5	4	0.79	0.80	0.95	0.44
	IUS and FC > 100	21	1	3	4	0.88	0.80	0.95	0.57

This table presents the diagnostic performance metrics for IUS, FC, and their combined use in predicting histologic remission in patients with ulcerative colitis, as reported by Sagami et al. [33] and Goodsall et al. [32]. For each diagnostic modality, the number of true positives (TPs), false positives (FPs), false negatives (FNs), and true negatives (TNs) is provided, alongside the corresponding sensitivity, specificity, positive predictive value (PPV), and negative predictive value (NPV).

**Table 4 ijms-26-08129-t004:** Pooled diagnostic accuracy estimates for intestinal ultrasound and faecal calprotectin in detecting histologic remission.

	Estimate	95% LCI	95% UCI
**Sensitivity**	0.839	0.351	0.980
**Specificity**	0.788	0.068	0.995
**DOR**	19.352	0.208	1803.149
**LR+**	3.959	0.175	89.568
**LR** **−**	0.205	0.026	1.624
**FPR**	0.212	0.005	0.932

This table summarizes the pooled diagnostic accuracy estimates derived from a meta-analysis evaluating the performance of IUS and FC in detecting histologic remission in patients with inflammatory bowel disease. The results include point estimates along with their corresponding 95% lower confidence interval (LCI) and upper confidence interval (UCI) values.

**Table 5 ijms-26-08129-t005:** Pooled diagnostic accuracy estimates for faecal calprotectin in detecting histologic remission.

	Estimate	95% LCI	95% UCI
**Sensitivity**	0.846	0.721	0.921
**Specificity**	0.600	0.380	0.786
**DOR**	8.250	2.562	26.569
**LR+**	2.115	1.222	3.663
**LR** **−**	0.256	0.123	0.533
**FPR**	0.400	0.214	0.620

This table summarizes the pooled diagnostic accuracy estimates for FC in detecting histologic remission in patients with inflammatory bowel disease, based on meta-analytic data. The table includes point estimates and their corresponding 95% lower (LCI) and upper (UCI) confidence intervals for each diagnostic parameter.

## Data Availability

The data used to support the findings of this study are available from the corresponding author upon request.

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
