# Peer review of "Non-Invasive vs. Invasive Markers in Ulcerative Colitis: A Systematic Review of Intestinal Ultrasound, Biopsy, and Faecal Calprotectin"

_ijms, 2025, doi:10.3390/ijms26178129_

Round 1

Reviewer 1 Report

Comments and Suggestions for Authors

Interesting paper indeed. Accurate assessment of histological remission is a critical prognostic goal in the management of ulcerative colitis (UC). Current routine evaluation is hindered by endoscopy and biopsy invasiveness. Non-invasive alternatives i.e., intestinal ultrasound (IUS) and Fecal Calprotectin (FC) promise for monitoring mucosal inflammation, though their ability to predict histological healing remains underexplored. Viviana Parra-Izquierdo et al. is a systematic review and meta-analysis aimed at evaluating the diagnostic accuracy of IUS, FC, and their combined use for detecting histological remission in patients with UC. A comprehensive literature search identified two eligible studies comprising 72 patients. Pooled estimates for IUS demonstrated high sensitivity but variable specificity, while FC alone exhibited high sensitivity with moderate specificity. Interestingly, and notably only one study assessed the combined prognostic disease activity approach, reporting superior performance with sensitivity and specificity and the certainty of the evidence rated as moderate. These findings suggest that a multimodal, non-invasive strategy combining IUS and FC may enhance prognostic accuracy for detecting histological remission in UC, potentially reducing the need for invasive procedures.

Monitoring disease activity involves assessing the severity and progression of a disease, often using a combination of clinical assessments, patient-reported outcomes, and laboratory or imaging tests. This process is crucial for guiding treatment decisions and improving patient outcomes, particularly in chronic conditions like UC, herein reviewed.

Note: Authors should define the difference between “diagnosis and prognosis.” Assessing the already diagnosed and known disease to monitor it severity, progress, and/or remission such as in UC herein reported is “Prognostic”. Often using a combination of clinical assessments, patient-reported outcomes, and laboratory or imaging tests. This is NOT Diagnostic but Prognostic. Please change the language to make it clear to leaders.

This is a well summarized review that is easy to follow and understand. It is genuinely nice work, and I agree with the authors that further large-scale and high-quality studies are warranted to validate these results and standardize prognostic protocols.

Comments on the Quality of English Language

Authors should define the difference between “Diagnosis and Prognosis.” Assessing the already diagnosed and known disease to monitor it severity, progress, and/or remission such as in UC herein reported is “Prognostic”. Often using a combination of clinical assessments, patient-reported outcomes, and laboratory or imaging tests. This is NOT Diagnostic but Prognostic. Please change the language to make it clear to leaders.

Author Response

Comment 1: Authors should define the difference between “diagnosis and prognosis.” Assessing the already diagnosed and known disease to monitor it severity, progress, and/or remission such as in UC herein reported is “Prognostic”. Often using a combination of clinical assessments, patient-reported outcomes, and laboratory or imaging tests. This is NOT Diagnostic but Prognostic. Please change the language to make it clear to leaders.
Response 1: Thank you for your valuable observation. We fully agree with your clarification regarding the distinction between diagnostic and prognostic terminology. As you correctly pointed out, the use of tools such as intestinal ultrasound and faecal calprotectin in patients with established ulcerative colitis primarily serves a prognostic purpose, namely, to monitor disease activity, histological remission, and therapeutic response, rather than to establish a new diagnosis.
In response, we have revised the manuscript to reflect this distinction more accurately. Throughout the text, we have replaced references to "diagnostic" accuracy or utility with "prognostic" accuracy or performance where appropriate, emphasizing the role of these tools in disease monitoring rather than initial diagnosis. We believe these changes will enhance clarity for readers and align with accepted definitions in the field.
Please see the highlighted lines 50-64, 146, 155-156, 520-523, 584-585, 628-630, and the recommendations section.
Thank you again for your insightful suggestion.

Comment 2: Comments on the Quality of English Language
Response 2: We appreciate the reviewer’s feedback regarding the quality of the English language used in the manuscript. We have undertaken a thorough review and revision of the entire text to enhance clarity, coherence, and grammatical accuracy.
To ensure a high standard of language, we have made the following improvements:
1. Grammar and Syntax: We have corrected grammatical errors and improved sentence structures to facilitate better comprehension.
2. Technical Language: We have refined technical terms and ensured that the scientific terminology is used consistently and appropriately throughout the manuscript.
3. Clarity and Precision: We have clarified ambiguous phrases and ensured that all descriptions are concise and precise.
Additionally, we sought assistance from a professional language editing service to further improve the quality of the manuscript. We believe that these revisions have significantly enhanced the overall readability and quality of the English language in the manuscript.

Reviewer 2 Report

Comments and Suggestions for Authors

This manuscript addresses a highly relevant clinical question regarding the accuracy of faecal calprotectin (FC) and intestinal ultrasound (IUS) in identifying histologic remission in ulcerative colitis (UC). The systematic review is methodologically sound, following Cochrane and PRISMA-DTA standards, and clearly reports pooled diagnostic parameters. Strengths include appropriate use of GRADE/QUADAS-2 tools, a thorough literature search, and a nuanced discussion on the clinical role of combined FC and IUS assessment.

However, there are a few points to consider:

1)While the authors have applied appropriate systematic review methodology, the inclusion of only two studies (each with distinct designs and diagnostic thresholds) limits the robustness of both qualitative synthesis and pooled estimates. The meta-analytical results, though technically valid, are highly imprecise & should be interpreted with great caution. I recommend revising the language in the abstract and conclusion to reflect the preliminary nature of the findings and the urgent need for further primary studies in this field. While this is acknowledged, conclusions in the abstract & discussion could be more cautious.

2)The rationale for choosing fixed-effect modeling for pooled estimates (when heterogeneity exists) should be better justified.

3)The report mentions one study using composite scores but does not attempt modeling. A brief note on why such modeling was not statistically feasible would have been helpful.

4)Although English is adequate and clear, minor revisions for flow (e.g., redundancy, sentence transitions) would improve clarity.

In summary, this work makes a valuable contribution to the literature on non-invasive UC monitoring and supports future multicenter prospective studies to validate the diagnostic synergy of FC and IUS.

Author Response

Comment 1: While the authors have applied, appropriate systematic review methodology, the inclusion of only two studies (each with distinct designs and diagnostic thresholds) limits the robustness of both qualitative synthesis and pooled estimates. The meta-analytical results, though technically valid, are highly imprecise & should be interpreted with great caution. I recommend revising the language in the abstract and conclusion to reflect the preliminary nature of the findings and the urgent need for further primary studies in this field. While this is acknowledged, conclusions in the abstract & discussion could be more cautious.

Response 1:
We sincerely thank the reviewer for this thoughtful and important observation. We fully acknowledge that the inclusion of only two eligible studies, each employing different methodologies and diagnostic thresholds, substantially limits the strength and generalizability of the pooled estimates.
In response to this comment, we have revised the language in both the abstract and discussion sections to highlight the preliminary nature of our findings clearly. We have reinforced the interpretation of the meta-analytic results as exploratory and imprecise due to the limited evidence base and significant heterogeneity. Additionally, we now explicitly state the need for further large-scale, methodologically rigorous primary studies to validate these findings and support the development of standardized prognostic protocols using non-invasive tools.
We appreciate this recommendation, which we believe enhances the scientific rigor and transparency of our manuscript.

Comment 2: The rationale for choosing fixed-effect modeling for pooled estimates (when heterogeneity exists) should be better justified
Response 2: Thank you for your insightful comment. We acknowledge that there was an error in the initial description of our meta-analytic approach. While the manuscript previously stated that a fixed-effect model was used, the analyses were conducted using a random-effects model, which is more appropriate given the expected heterogeneity in study populations, designs, and thresholds. We have corrected this statement in the Methods section and clarified the rationale for selecting a random-effects model. Specifically, we now explicitly describe that the random-effects model accounts for both within- and between-study variability. That heterogeneity was assessed using the I² statistic and visual inspection of forest plots and SROC curves.
We appreciate your careful reading and thank you for helping us improve the accuracy and clarity of the manuscript. Please see the highlighted lines 320, 322-329.

Comment 3: The report mentions one study using composite scores but does not attempt modeling. A brief note on why such modeling was not statistically feasible would have been helpful.
Response 3: Thank you for this thoughtful observation. We agree that modeling the diagnostic performance of the composite IUS + FC approach could have strengthened the analysis. However, such modeling was not statistically feasible due to the limited number of included studies. Only one study (Goodsall et al., 2024) provided data on the combined use of IUS and FC. Without at least two independent datasets reporting composite results, pooled estimates or meta-analytic modeling could not be performed reliably. We have now clarified this limitation in both the Methods and Discussion sections of the manuscript to reflect the statistical constraints and the need for further studies reporting composite diagnostic data. Please see the highlighted lines 329-333, 636-637 and 645-658
We appreciate your suggestion, which has helped us improve the transparency and completeness of the manuscript.

Comment 4: Although English is adequate and clear, minor revisions for flow (e.g., redundancy, sentence transitions) would improve clarity.
Response 4: We appreciate the reviewer’s feedback regarding the quality of the English language used in the manuscript. We have undertaken a thorough review and revision of the entire text to enhance clarity, coherence, and grammatical accuracy.
To ensure a high standard of language, we have made the following improvements:
1. Grammar and Syntax: We have corrected grammatical errors and improved sentence structures to facilitate better comprehension.
2. Technical Language: We have refined technical terms and ensured that the scientific terminology is used consistently and appropriately throughout the manuscript.
3. Clarity and Precision: We have clarified ambiguous phrases and ensured that all descriptions are concise and precise.
Additionally, we sought assistance from a professional language editing service to further improve the quality of the manuscript. We believe that these revisions have significantly enhanced the overall readability and quality of the English language in the manuscript.